Changes in the timing of departure and arrival of Irish migrant waterbirds

Donnelly Alison 1 alison.c.donnelly@gmail.com
Geyer Heather 2
Yu Rong 3
1 Department of Geography, University of Wisconsin-Milwaukee , WI , USA
2 Centre for the Environment, Trinity College Dublin , Dublin , Ireland
3 Fulton Schools of Engineering, Arizona State University , Mesa, AZ , USA
Gandini Patricia
Electronic publication date: 2015 Jan 15
Publication date: 2015
Volume: 3
Electronic Location ID: e726
Received 2014 Sep 29; Accepted 2014 Dec 24
Copyright: © 2015 Donnelly et al.
Copyright year: 2015
Copyright holder: Donnelly et al.
License: This is an open access article distributed under the terms of the Creative Commons Attribution License, which permits unrestricted use, distribution, reproduction and adaptation in any medium and for any purpose provided that it is properly attributed. For attribution, the original author(s), title, publication source (PeerJ) and either DOI or URL of the article must be cited.
License URL: https://creativecommons.org/licenses/by/4.0/

Keywords: Spring arrival and departure, Temperature, Ireland, Waterbirds

Funding: Irish Environmental Protection Agency 2007-CCRP-2.4 This work was in part funded by the Irish Environmental Protection Agency under the STRIVE programme, project number 2007-CCRP-2.4. The funders had no role in study design, data collection and analysis, decision to publish, or preparation of the manuscript.

==============================
There have been many recent reports across Europe and North America of a change in the timing of arrival and departure of a range of migrant bird species to their breeding grounds. These studies have focused primarily on passerine birds and climate warming has been found to be one of the main drivers of earlier arrival and departure in spring. In Ireland, rising spring temperature has been shown to result in the earlier arrival of sub-Saharan passerine species and the early departure of the Whooper Swan. In order to investigate changes in spring arrival and departure dates of waterbirds to Ireland, we extracted latest dates as an indicator of the timing of departure of winter visitors (24 species) and earliest dates as an indicator of the timing of arrival of spring/summer migrants (2 species) from BirdWatch Ireland’s East Coast Bird reports (1980–2003). Three of the winter visitors showed evidence of later departure and one of earlier departure whereas one of the spring/summer visitors showed evidence of earlier arrival. In order to determine any influence of local temperature on these trends, we analysed data from two synoptic weather stations within the study area and found that spring (average February, March and April) air temperature significantly (P < 0.05) increased at a rate of 0.03 °C per year, which was strongly correlated with changes in latest and earliest records. We also tested the sensitivity of bird departure/arrival to temperature and found that Northern Pintail would leave 10 days earlier in response to a 1 °C increase in spring temperature. In addition, we investigated the impact of a large-scale circulation pattern, the North Atlantic Oscillation (NAO), on the timing of arrival and departure which correlated with both advances and delays in departure and arrival. We conclude that the impact of climate change on earliest and latest records of these birds is, as expected, species specific and that local temperature had less of an influence than large-scale circulation patterns.

Introduction

Despite the vast amount of research being carried out on climate change, there remains a degree of uncertainty as to the nature of any future impacts on ecosystems (IPCC, 2013). This uncertainty stems from the range of complexities associated with climate change coupled with the fact that changes are expected to be region-specific (Alcamo et al., 2007). While the exact nature of future climate remains unclear, it is now understood that the changes that have occurred are outside the realm of natural variability (Hurrell & Trenberth, 2010; IPCC, 2007; IPCC, 2013). Understanding the environmental implications which have occurred to date as a result of this anthropogenically forced climate change will provide better awareness of what future impacts to expect. Furthermore, the influence of large-scale oscillations in climate, such as the North Atlantic Oscillation (NAO) which also impact spring temperature should also be considered when examining the influence of climate on wildlife.

The North Atlantic Oscillation is a large-scale fluctuation in atmospheric mass between high and low pressure cells which are centred near the Azores and Iceland (Cook, Smith & Mann, 2005). It is considered to be the most significant atmospheric oscillation in the North Atlantic region (Stenseth et al., 2003; Goodkin et al., 2008), with its climatic effects stretching east to the United States, north to Greenland, south to Africa and west into Asia. Although active throughout the whole of the year, its effects are strongest from November to April (Visbeck et al., 2001; Goodkin et al., 2008; Stenseth et al., 2003). A positive NAO is associated with warmer winter temperatures along with stronger westerly winds across the North Atlantic. A negative NAO is associated with cooler winter temperatures (Hurrell, 1995; Cook, Smith & Mann, 2005). In addition to interannual variability, multidecadal patterns of the NAO have also been revealed. Recent research suggests that these patterns have become increasingly pronounced, particularly over the last half of the 20th century (Hurrell & Van Loon, 1997; Goodkin et al., 2008) and suggest that anthropogenic forcing may be responsible for the intensified nature of the more recent NAO phases (Goodkin et al., 2008). Increasing global temperatures coupled with increasingly exaggerated NAO phases makes predicting the impact of climate change on ecosystems extremely challenging.

As a means of quantifying the impact of climate change on wildlife, much research has focused on shifts in phenology. These recurring annual events, including bird migration, are often strongly correlated with temperature Cotton, 2003; Jonzén, Lindén & Ergon, 2006; Sparks et al., 2005; Miller-Rushing et al., 2008; Donnelly et al., 2009. Migration has evolved in a number of species as an adaptive strategy to seasonal changes in the abundance of essential resources (Berthold, 2001). It is an energetically expensive and potentially hazardous adaptation in which individuals must cope with a variety of changes including shifts in food availability, weather and social behaviour (O’Reilly & Wingfield, 1995).

While migration has enabled birds to utilize typically unsuitable habitats, it also increases their vulnerability to climatic variability. Weather influences their foraging behaviour, metabolic rate, and breeding success (Crick, 2004). Since climate change is having a significant impact on both the structure and the functioning of ecosystems, it is highly likely that some migratory waterbirds may be affected by, for example, changes in the timing of resource availability at breeding grounds. In the past century, Europe has experienced an increase in annual average air temperature of approximately 0.8 °C (IPCC, 2013). This increase has been linked to the earlier nesting dates as well as the earlier arrival and departure of some migrant bird species (Cotton, 2003; Jonzén, Lindén & Ergon, 2006; Crick, 2004; Donnelly et al., 2009; Stirnemann et al., 2012). While an advance in the timing of first arrival dates has often been reported as an indicator of warming, care must be exercised not to interpret or extrapolate these trends to the population level (Sparks, Roberts & Crick, 2001). The authors argue that data on the complete migration distribution may often not be available, therefore reporting a change in one tail of a distribution is nonetheless valid, if not ideal.

A study focusing on the timing of spring migrant passerine birds to Ireland observed a trend of earlier arrival in long-distance migrants during the spring, which was found to be directly related to increasing air temperatures (Donnelly et al., 2009). In addition, a winter visitor, the Whooper Swan has also been reported to leave earlier in spring in response to rising spring temperature (Stirnemann et al., 2012). Research conducted by Rubolini et al. (2007) suggests that the response of migratory birds to changing climatic conditions is species specific. They recommend that phylogenetic history be taken into account when comparing those responses, as closely related species are more likely to show similarities. The study also indicated that spatial variability and migratory patterns influence response to warming, with short-distance migrants advancing their spring arrival dates more so than long-distance migrants, although other studies albeit using different species and geographical location suggest otherwise, emphasizing the complex nature of migratory patterns and phenotypic plasticity (Rubolini et al., 2007; Jonzén, Lindén & Ergon, 2006).

As Ireland lies directly within the East Atlantic Flyway and remains an important stopover point and final destination for migratory bird species, understanding the impact of climate change on bird migration is important, especially in relation to the European Union’s Habitat and Birds Directives (UNEP and CMS, 2011). Recent studies have reported an advance in the timing of both leafing of trees (Donnelly, Salamin & Jones, 2006) and arrival of spring migrant birds (Donnelly et al., 2009) in response to warming, thus demonstrating a detectable impact of climate change on Irish wildlife. Future estimates predict that annual mean temperature in Europe will likely surpass the global mean (IPCC, 2013). These unprecedented increases in temperature will continue to have knock-on effects throughout Irish ecosystems.

In this study, annual bird reports covering the east coast region of Ireland were utilized to create a unique dataset of latest recorded dates in spring for winter visiting waterbirds and earliest dates for spring/summer visitors as indicators for departure (winter visitors) and arrival (spring/summer arrivals). In addition, local spring air temperature and trends in the North Atlantic Oscillation Index for the time period of interest were also obtained. Together these datasets were used to (i) investigate if changes in the timing of spring departure/arrival of migrant waterbirds occurred over the time period (1980–2003) and (ii) determine whether or not local temperature and/or large-scale short-term circulation patterns were driving any observed phenological trend.

Materials and Methods

Bird phenological data

In order to examine arrival and departure dates of migrant waterbirds to Ireland, we examined records from the Irish East Coast Bird Reports (BirdWatch Ireland) over a 24-year period (1980–2003). These reports were compiled from records submitted by known birdwatchers to BirdWatch Ireland of sightings and counts of all bird species within part of the East coast region (Fig. 1). We extracted earliest reporter dates in spring as an indicator of the timing of arrival of spring/summer migrants and latest reported dates as an indicator of the timing of departure of winter visitors.

Figure 1 Location of the East coast region of Ireland (includes the counties, from North to South, Louth, Meath, Dublin and Wicklow) where the bird data were recorded and the two synoptic weather stations (T1: Dublin Airport and T2: Casement Aerodrome) used for temperature data.

The Irish East Coast Bird Reports (Cooney et al., 1980; Cooney et al., 1981; Cooney et al., 1982; Cooney et al., 1983; Cooney et al., 1984; Cooney et al., 1985; Cooney, Madden & O’Flanagan, 1986; Cooney et al., 1987; Cooney et al., 1988; Cooney, Madden & O’Donnell, 1989; Cooney, Madden & O’Donnell, 1990; Cooney, Madden & O’Donnell, 1991; Cooney & Madden, 1992; Cooney & Madden, 1993; Cooney & Madden, 1994; Cooney & Madden, 1995; Madden & Cooney, 1996; Madden, 1998; Madden & Cooney, 1999; Coombes & Murphy, 2000; Coombes & Murphy, 2001; Coombes & Murphy, 2002; Coombes & Murphy, 2003) have been compiled and published by BirdWatch Ireland (formerly known as the Irish Wildbird Conservancy) from 1980 to 2003, with the exception of 1997 when funding was unavailable. A full set of the reports is available from the library at BirdWatch Ireland (www.birdwatchireland.ie) and at two other libraries that we know of: Trinity College Dublin (www.tcd.ie), Ireland and University College Dublin (www.ucd.ie), Ireland. The approximate length of the coastal area covered by the reports is 200 km and spans four counties: Louth, Meath, Dublin and Wicklow (Fig. 1). The reports were edited by between 1 and 5 professionals working for BirdWatch Ireland who validated each record prior to publication. Contributors to the reports included both professional and amateur birdwatchers and ranged in number from 43 to 108, with an average of 76 contributors per year. The list of contributors is provided in each report and many of the contributors’ names are consistent over the time period of the reports. It was not possible to accurately ascertain the frequency at which observations were made as this level of detail was not provided.

The focus of this study was specifically on early and late dates of migrant birds as an indication of possible changes in the timing of spring arrival and departure, but we were nonetheless cognisant of using records of one individual bird as representing the population as a whole. Therefore, when the recorded number of individuals in a particular species was consistently high (e.g., Red Knot) over the time period, records of <8–10 individuals were not included in the statistical analysis, whereas when numbers were typically low (e.g., Long-tailed Duck) records of 2–3 individuals were included. Data that were excluded from analysis, as numbers were considered misrepresentative, were as follows; Red-throated Diver (2 individuals reported on DOY 210 in 1992), Great Northern Diver (singles up to DOY 189 in 1984), Greylag Goose (1 individual on DOY 143 in 1981), Brent Goose (2 individuals on DOY 161 in 1984 and 1 on DOY 175 in 1990), Eurasian Wigeon (1 on DOY 173 in 1981, 1 on DOY 158 in 1982 and 2 on DOY 167 in 1984), Eurasian Teal (1 on DOY 197 in 1991), Common Pochard (2 on unspecified dates in Jun., Jul. and Aug. 1981), Greater Scaup (1 on DOY 156 in 1985), Common Goldeneye (1 on DOY 137 in 1987), Grey Plover (1 on DOY 146 in 1987) and Common Greenshank (1 on DOY 165 in 1986). In addition, when records of a particular bird were available for each month of the year, making it impossible to extract latest or earliest dates, data for these years were excluded. For example, in 1987 Sanderling were recorded in every month of the year and so data for this year were omitted. Similarly, data for Common Scoter (1985), Grey Plover (1989), Purple Sandpiper (1989), Black-tailed Godwit (1986–1988), Common Redshank (1991), Common Greenshank (1991) and Ruddy Turnstone (1986–1989) were also omitted from the analysis. The average number of birds (recorded on the day observations were being made) per year is reported in Table 1 together with the standard deviation to present the large inter-species and interannual variation in numbers.

Table 1 Common and scientific names of the bird species together with category, diver, waterfowl or wader; ‘N Year’ number of records in the time series; ‘N Bird’ average number of birds reported per year together with the Standard Deviation; MK-Stat results of the Mann–Kendall test to determine trends over time and significance level (P); numbers in bold are statistically significant. Earliest, average and latest DOY (day of year) refers to the data extracted from the Irish East Coast Bird Reports (1980–2003) for each individual species.

Species	Category	N Year	N Bird	MK-Stat	P-value	Earliest DOY	Average DOY	Latest DOY	
Winter visitors	
Red-throated Diver Gavia stellata	Diver	22	27 ± 39	24	0.4976	49	103	124	
Great Northern Diver Gavia immer	Diver	17	8 ± 9	30	0.2165	52	106	163	
Whooper Swan Cygnus cygnus	Waterfowl	23	20 ± 29	−3	0.9367	73	101	205	
Greylag Goose Anser anser	Waterfowl	23	148 ± 203	22	0.5611	60	96	121	
Brent Goose Branta bernicla	Waterfowl	18	161 ± 281	25	0.3437	60	108	177	
Eurasian Wigeon Anas Penelope	Waterfowl	17	194 ± 236	41	0.0864	52	83	168	
Eurasian Teal Anas crecca	Waterfowl	16	121 ± 53	11	0.6191	24	73	96	
Northern Pintail Anas acuta	Waterfowl	15	70 ± 100	42	0.0374	33	59	85	
Common Pochard Aytha farina	Waterfowl	18	107 ± 161	11	0.6765	37	78	186	
Greater Scaup Aytha marila	Waterfowl	19	59 ± 138	1	0.9721	39	93	177	
Long-tailed Duck Clangula hyemalis	Waterfowl	22	7 ± 10	−5	0.8879	33	78	128	
Common Scoter Melanitta nigra	Waterfowl	18	331 ± 531	−3	0.9094	45	90	165	
Common Goldeneye Bucephala clangula	Waterfowl	16	58 ± 86	1	0.9641	40	85	200	
Grey Plover Pluvialis squatarola	Wader	16	167 ± 131	−5	0.8214	37	86	194	
Red Knot Calidris canutus	Wader	16	1054 ± 1757	7	0.7524	49	82	141	
Sanderling Calidris alba	Wader	17	147 ± 114	6	0.8048	60	123	210	
Purple Sandpiper Calidris maritima	Wader	21	20 ± 25	80	0.0156	19	122	212	
Dunlin Calidris alpina	Wader	17	1429 ± 1497	61	0.0118	52	95	147	
Jack Snipe Lymncryptes minimus	Wader	20	2 ± 3	−35	0.2559	29	70	104	
Black-tailed Godwit Limosa limosa	Wader	19	195 ± 229	−49	0.0861	86	146	196	
Bar-tailed Godwit Limosa lapponica	Wader	15	326 ± 281	−25	0.2160	52	114	201	
Common Redshank Tringa tetanus	Wader	18	558 ± 350	3	0.9094	43	84	111	
Common Greenshank Tringa nebularia	Wader	15	15 ± 12	−47	0.0200	32	106	196	
Ruddy Turnstone Arenaria interpres	Wader	15	91 ± 68	29	0.1513	60	119	182	
Summer visitors	
Manx Shearwater Puffinus puffinus	Diver	20	626 ± 1801	−16	0.6028	78	124	163	
Sandwich Tern Sterna sandvicensis	Diver	22	2 ± 2	−117	0.0009	71	85	103	

Species considered for this study were categorized as follows: divers (aquatic diving bird e.g., Red-throated Diver, Sandwich Tern), waterfowl (ducks and geese e.g., Wigeon) and waders (forage primarily along the seashore e.g., Grey Plover) for which earliest and latest dates were published in the reports (Table 1). Since there were many instances of missing values, only species with a minimum of 15 years of records were included, as this was considered sufficient for robust statistical analysis (following Rubolini et al., 2007; Donnelly et al., 2009). All earliest and latest dates were converted to day of year (DOY). Statistical analyses were performed on 26 species of migratory waterbirds out of a total of 40 species considered. The following 14 species did not meet the required criteria: Garganey Anas querquedula Eider Somateria mollissima; Shoveler Anas clypeata; Goldeneye Bucephala clangula; Goosander Mergus merganser; Velvet Scoter Mellanitta fusca; Ruddy Duck Oxyura jamaicensis; Spotted Redshank Tringa erythropus; Green Sandpiper Tringa ochropus; Iceland gull Larus glaucoides; Arctic tern Sterna paradisaea; Roseate Tern Sterna dougallii; Little Tern Sterna albifrons; Puffin Fratercula arctica.

Climate data

The climate data used in this study were supplied by the Irish meteorological service (http://www.met.ie/climate/climate-data-information.asp), Met Éireann. In order to examine the impact of local temperature on the timing of earliest and latest dates of migrant waterbirds, the average monthly air temperatures for the study area were acquired, using data from the service’s two synoptic weather stations located at Dublin airport (53°25′40″N, 6°14′27″W) and Casement aerodrome (53°18′20″N, 6°26′20″W) (Fig. 1). The furthest coastal point from the nearest weather station was approximately 50 km. Since the monthly average temperature difference between the sites was consistently less than 0.5 °C, an average for the study area was calculated and used in all subsequent analyses. Daily maximum and minimum temperatures were averaged for the months of January, February, March and April. In addition, average temperatures were also calculated from February to April, representing ‘average spring temperatures.’

Furthermore, the influence of large-scale circulation patterns on the timing of arrival was investigated. Data for the winter (December–February) North Atlantic Oscillation (NAO) were obtained from the Climate Research Unit websites at East Anglia University, Norwich, UK for years 1980–2000 (http://www.cru.uea.ac.uk/cru/data/nao.htm/) and for 2001–2003 (http://www.cru.uea.ac.uk/~timo/datapages/naoi.htm).

Statistical analysis

The latest date (Day of Year) recorded in spring of 24 winter species and the earliest date of two spring/summer species (Table 1 and Fig. 2) of migrant waterbird was first plotted against year (1980–2003) to determine any temporal pattern in the time series (e.g., Fig. 2). Trends in the timing of earliest and latest dates (DOY) were analyzed independently for each species using the MULTMK/PARTMK program for the computation of univariate Mann–Kendall tests (Libiseller & Grimvall, 2002). The Mann–Kendall test is a robust method for examining monotonic trends in a time series, has the advantage of being able to deal with missing values (Wahlin & Grimcall, 2010) and has been previously used in similar studies (Richardson et al., 2006; Donnelly et al., 2009).

Figure 2 Average spring (February–April) temperature on the east coast of Ireland for the period 1980–2003 showing a statistically significant increasing trend.

Linear regression was used to determine the trend in average spring temperature and to examine the relationship between departure/arrival and the climatic variables, similar to Miller-Rushing et al. (2008). We first tested for temporal autocorrelation in departure/arrival and climate parameters (including NAOI) using the Box-Ljung test and found only 3 instances of significant lag-1 autocorrelation: Pochard (correlation for lag-1 = 0.389 ± 0.019, Box-Ljung statistic = 4.181, P = 0.04), Knot (correlation for lag-1 = 0.403 ± 0.0195, Box-Ljung statistic = 4.252, P = 0.04) and Tern (correlation for lag-1 = 0.410 ± 0.019, Box-Ljung statistic = 4.668, P = 0.03). We detrended these data by subtracting the value in year (i) from that in year (i−1) and repeated the multiple regression analysis. Since neither the original data nor the detrended data showed significant correlations with temperature variables and NAOI, we used the original data for simplicity and consistency (following Balbontín et al., 2009). All statistical tests (apart from the Mann–Kendall test) were carried out using IBM SPSS version 20.0.

Results

Temperature trends

As expected, monthly average spring (February to April) air temperatures, averaged across two meteorological stations (Dublin Airport and Casement Aerodrome) within the study area for the period 1980 to 2003 showed a statistically significant (R2 = 0.193; P = 0.0541) increasing trend (0.04 °C per year) towards warmer temperatures (Fig. 2).

Temporal pattern in departure and arrival of migrant waterbird species

The number of individuals recorded on the latest (winter visitors) and earliest (spring visitors) dates in spring is presented in Table 1 (note: these data do not represent peak counts but rather the numbers present on the particular date). The largest numbers reported were waders such as Red Knot (1,054 ± 1,757), Dunlin (1,429 ± 1,497) and Common Redshank (580 ± 350), whereas smaller numbers tended to be divers and waterfowl such as Great Northern Diver (8 ± 9) and Long-tailed Duck (7 ± 10) although one species of wader (Jack Snipe) was also reported in small numbers (2 ± 3). Numbers reported on the earliest spring dates of one of the two summer visitors were large (Manx Shearwater 626 ± 1,801) whereas a similar result for the Sandwich Tern was 2 ± 2. These data reflect variation between species in numbers of both early and late recordings.

One waterfowl and two waders (Northern Pintail (P = 0.0374), Purple Sandpiper (P = 0.0156) and Dunlin (Fig. 3A P = 0.0118)) revealed a statistically significant later trend in the latest recorded date in spring over the time period examined (Table 1). Even though the majority (16 out of 24) of species showed a tendency to remain in Ireland longer, as indicated by positive values resulting from the Mann–Kendall test, most were not statistically significant. The latest recorded dates of one wader (the Common Greenshank (P = 0.0200)) out of the eight species exhibiting negative M–K trends was statistically significantly earlier. The earliest observed dates of one of the two spring/summer visiting waterbirds (Sandwich Tern (Fig. 3B P = 0.0009)) became earlier over time.

Figure 3 Trends in the timing of (a) the latest recorded date in spring (indicator for departure) when Dunlin, a winter visitor, was observed and (b) the earliest recorded date in spring (indicator for arrival) when Sandwich Tern, a spring/summer visitor, was observed in the east coast region of Ireland over the time period 1980–2003.

Overall the earliest date by which the wintering birds left the east coast of Ireland was DOY 19 (19 January) whereas the latest date was DOY 212 (31 July), reflecting the wide range of latest dates recorded between 1980 and 2003 (Table 1). The earliest dates the two spring/summer migrants arrived was DOY 71 (12 March) and the latest was DOY 163 (12 June), which reflects a much narrower time period but there were only 2 species of waterbird examined in this category: the Manx Shearwater and the Sandwich Tern.

Spring temperature as a driver of early migration phenology

The influence of local spring temperature on latest (winter visitors) and earliest (spring/summer visitors) bird records revealed both positive and negative trends depending on the species in question (Table 2). Of the months considered, temperature in March showed the strongest correlations. The timing of departure (latest records) of only 1 of the twenty-four winter visitors examined showed positive correlations with March temperature (Whooper Swan (12d °C−1; R2 = 0.182; P = 0.042)). This suggests that when March temperature was relatively warm this species remained in Ireland later into the season. However, some species showed the opposite trend. For example, the timing of departure of both Red-throated Diver (−6d °C−1; R2 = 0.173; P = 0.054) and Northern Pintail (−8d °C−1; R2 = 0.378; P = 0.019) became significantly earlier as March temperature increased, suggesting that these waterbirds leave Ireland earlier as March temperature rises (Table 2). The earliest records observed of the spring/summer visitors also became progressively earlier in the season as March temperature increased but this trend was not statistically significant (Table 2).

Table 2 Linear regression slopes ±SE, R2 and P values for latest (winter visitors) records (DOY) and earliest (summer visitors) records (DOY) of migrant waterbirds to the east coast of Ireland and (i) average monthly local spring temperatures (January–April; and average February to April) (°C) and (ii) the North Atlantic Oscillation Index (NAOI).

Species	January	February	March	April	F–A	NAOI	
	Slope ±
SE	R 2	P	Slope ±
SE	R 2	P	Slope ±
SE	R 2	P	Slope ±
SE	R 2	P	Slope ±
SE	R 2	P	Slope ±
SE	R 2	P	
Winter visitors	
Red-throated
Diver	−4.245 ±
2.99	0.092	0.170	−2.935 ±
2.63	0.059	0.278	−6.344 ±
3.10	0.173	0.054	−0.690 ±
3.87	0.002	0.860	−7.816 ±
4.54	0.129	0.101	−9.030 ±
2.54	0.386	0.002	
Great
Northern
Diver	0.410 ±
7.14	0.000	0.955	5.307 ±
5.47	0.059	0.347	0.705 ±
7.36	0.001	0.925	9.384 ±
8.73	0.071	0.300	10.554 ±
10.01	0.069	0.309	1.245 ±
7.06	0.002	0.862	
Whooper
Swan	0.191 ±
5.46	0.000	0.972	−4.029 ±
4.55	0.036	0.385	11.881 ±
5.50	0.182	0.042	−3.615 ±
6.37	0.015	0.576	1.059 ±
8.28	0.001	0.899	−0.692 ±
5.74	0.001	0.905	
Greylag
Goose	4.630 ±
2.66	0.126	0.097	−1.316 ±
2.40	0.014	0.589	5.287 ±
2.95	0.133	0.088	−2.714 ±
3.30	0.031	0.420	0.366 ±
4.32	0.000	0.933	4.501 ±
2.83	0.108	0.126	
Brent
Goose	−3.061 ±
7.37	0.011	0.683	−6.933 ±
5.87	0.080	0.254	−7.203 ±
6.77	0.066	0.303	1.326 ±
7.90	0.002	0.869	−13.863 ±
11.04	0.090	0.227	−9.399 ±
6.121	0.128	0.144	
Eurasian
Wigeon	−0.714 ±
6.54	0.001	0.915	−7.670 ±
4.75	0.148	0.127	−0.355 ±
6.99	0.000	0.960	−13.018 ±
7.01	0.187	0.083	−16.462 ±
8.85	0.187	0.083	−0.084 ±
6.26	0.000	0.990	
Eurasian
Teal	0.918 ±
3.94	0.004	0.819	−3.916 ±
3.02	0.107	0.216	1.101 ±
4.42	0.066	0.807	−4.181 ±
4.88	0.50	0.406	−5.607 ±
5.64	0.257	0.337	−1.510 ±
3.81	0.105	0.698	
Northern
Pintail	−0.649 ±
3.89	0.002	0.870	−5.916 ±
2.13	0.392	0.017	−8.049 ±
2.98	0.378	0.019	0.130 ±
4.20	0.000	0.976	−10.443 ±
3.75	0.392	0.017	−6.293 ±
3.23	0.240	0.075	
Common
Pochard	8.743 ±
8.24	0.066	0.304	4.961 ±
6.52	0.035	0.458	8.571 ±
8.44	0.061	0.325	9.069 ±
10.27	0.047	0.390	14.173 ±
11.21	0.091	0.224	11.549 ±
9.03	0.093	0.219	
Greater
Scaup	7.638 ±
8.10	0.050	0.359	6.396 ±
6.82	0.049	0.361	10.266 ±
9.08	0.070	0.274	−1.138 ±
10.42	0.001	0.914	13.660 ±
12.75	0.063	0.299	18.328 ±
7.96	0.238	0.034	
Long-tailed
Duck	3.883 ±
3.77	0.050	0.317	−5.488 ±
3.13	0.133	0.095	−0.611 ±
4.23	0.001	0.887	−6.035 ±
4.62	0.079	0.206	−9.477 ±
5.65	0.123	0.109	−2.240 ±
4.00	0.015	0.582	
Common
Scoter	−5.834 ±
6.22	0.052	0.362	7.072 ±
4.81	0.003	0.827	3.048 ±
6.97	0.012	0.668	0.725 ±
6.66	0.001	0.915	3.257 ±
08.82	0.008	0.717	1.872 ±
6.41	0.005	0.774	
Common
Goldeneye	6.724 ±
11.28	0.025	0.561	1.608 ±
8.03	0.003	0.844	3.546 ±
10.73	0.008	0.746	−0.408 ±
12.52	0.000	0.974	4.179 ±
15.36	0.005	0.790	22.60 ±
9.62	0.283	0.034	
Grey Plover	−19.623 ±
9.50	0.233	0.054	−10.515 ±
7.20	0.132	0.166	−14.105 ±
10.42	0.116	0.197	−8.181 ±
11.61	0.034	0.493	−25.017 ±
13.04	0.208	0.076	−17.040 ±
10.13	0.168	0.115	
Red
Knot	4.673 ±
5.75	0.044	0.433	1.441 ±
4.39	0.008	0.747	5.180 ±
5.50	0.060	0.362	6.226 ±
7.25	0.050	0.405	7.289 ±
7.68	0.060	0.359	4.490 ±
5.37	0.048	0.417	
Sanderling	1.141 ±
8.67	0.001	0.897	−8.551 ±
6.23	0.112	0.190	−6.668 ±
9.71	0.030	0.503	−2.569 ±
10.38	0.004	0.808	−14.185 ±
11.59	0.091	0.240	−10.091 ±
7.88	0.098	0.220	
Purple
Sandpiper	11.659 ±
7.83	0.105	0.153	6.429 ±
6.46	0.050	0.332	5.413 ±
8.59	0.020	0.536	12.690 ±
9.41	0.087	0.193	16.499 ±
11.29	0.101	0.160	11.111 ±
8.09	0.090	0.186	
Dunlin	6.206 ±
7.62	0.042	0.428	5.414 ±
5.66	0.057	0.354	4.283 ±
7.82	0.020	0.592	2.816 ±
9.19	0.006	0.764	10.227 ±
10.52	0.059	0.346	−0.227 ±
7.54	0.000	0.976	
Jack
Snipe	−3.104 ±
4.05	0.032	0.454	−2.269 ±
3.11	0.029	0.474	−0.687 ±
4.22	0.001	0.873	−1.299 ±
4.40	0.005	0.771	−3.485 ±
5.54	0.022	0.537	−6.848 ±
3.70	0.160	0.081	
Black-tailed
Godwit	−4.024 ±
5.84	0.027	0.500	−6.126 ±
5.69	0.064	0.297	0.081 ±
6.64	0.000	0.990	−5.269 ±
8.20	0.024	0.529	−10.475 ±
10.88	0.052	0.349	−3.841 ±
6.05	0.023	0.534	
Bar-tailed
Godwit	−12.922 ±
11.47	0.089	0.280	−1.287 ±
10.80	0.001	0.907	3.437 ±
13.44	0.005	0.802	−4.519 ±
15.52	0.006	0.776	−1.796 ±
22.40	0.000	0.937	3.403 ±
11.28	0.007	0.768	
Common
Redshank	−1.548 ±
3.95	0.010	0.700	1.630 ±
3.09	0.017	0.605	−0.688 ±
3.95	0.002	0.864	3.457 ±
4.43	0.037	0.447	2.806 ±
5.31	0.017	0.605	2.571 ±
3.75	0.029	0.502	
Common
Greenshank	2.872 ±
11.47	0.005	0.806	−8.566 ±
8.76	0.069	0.346	11.622 ±
11.78	0.070	0.342	−1.477 ±
13.47	0.001	0.914	−3.062 ±
16.38	0.003	0.855	−3.249 ±
10.83	0.007	0.769	
Ruddy
Turnstone	1.669 ±
8.35	0.003	0.845	−10.354 ±
6.25	0.174	0.122	−4.142 ±
8.32	0.019	0.627	−2.134 ±
10.41	0.003	0.841	−18.942 ±
12.80	0.144	0.163	−5.837 ±
7.37	0.046	0.443	
Summer visitors	
Manx
Shearwater	−1.675 ±
4.31	0.008	0.702	−0.616 ±
3.81	0.001	0.874	−5.672 ±
4.75	0.073	0.248	4.317 ±
5.20	0.037	0.417	−1.838 ±
6.74	0.004	0.788	−1.753 ±
4.64	0.008	0.710	
Sandwich
Tern	−1.359 ±
1.57	0.036	0.396	0.937 ±
1.29	0.026	0.477	−0.617 ±
1.78	0.006	0.732	0.884 ±
1.89	0.011	0.644	1.084 ±
2.32	0.011	0.646	1.156 ±
1.63	0.025	0.486	

Sensitivity to overall spring warming

The response of the timing of departure and arrival to a 1 °C increase in average spring temperature varied greatly between species (Table 2) indicating that some birds showed greater sensitivity to changing temperature than others. The Northern Pintail (−10d °C−1; R2 = 0.392; P = 0.017; Fig. 4) significantly advanced their timing of departure in response to a 1 °C increase in average spring temperature, whereas none of the positive trends were statistically significant. The summer visitors showed only weak correlations with temperature variables.

Figure 4 Trend in the timing of departure (latest spring record) of Northern Pintail to the east coast of Ireland over the time period 1980–2003 in relation to average spring (February–April) temperature.

Influence of large-scale circulation patterns on bird migration

The timing of departure (latest spring record) of the Red-throated Diver (−9d index unit-1; R2 = 0.386; P = 0.002; Fig. 5) was found to be statistically negatively correlated with the winter (December–February) NAOI (Table 2). This negative correlation suggests that during a positive NAO phase the timing of departure was earlier in the year (Fig. 5). Two (Greater Scaup (18d index unit-1; R2 = 0.386; P = 0.002) and Common Goldeneye (23d index unit-1; R2 = 0.386; P = 0.002)) of the 10 species exhibiting positive correlations were statistically significant, suggesting that departure was later when the NAOI was positive (Table 2). Both summer visitors showed only weak correlations with the NAO index.

Figure 5 Trend in the timing of the latest spring record of Red-throated Diver in the east coast region of Ireland over the time period 1980–2003 in relation to the North Atlantic Oscillation—winter index.

Discussion

Our results indicate a change in the timing of the latest and earliest recorded dates in spring of a number (5 out of 26) of migrant waterbirds to the east coast region of Ireland over a 24-year period (1980–2003). We proposed that the latest recorded date in spring of winter visitors may be used as an indicator of the timing of last departure, whereas the earliest recorded date in spring of spring/summer migrants may be used as an indicator of the timing of first arrival. Even though the records reported in BirdWatch Ireland’s East Coast Bird Reports were not specifically collected for the purpose of determining latest departure or earliest arrival dates of migrant birds, given the systematic and rigorous manner used to generate these data we are confident that the trends reported here are real. However, more detailed records, such as daily counts, would undoubtedly give a more accurate account of migration at a population level, but in the absence of such data these reports yielded some interesting trends. Furthermore, keeping in mind Aristotle’s phrase ‘one swallow does not make a summer,’ we attempted to avoid incorporating outliers as representative of wider behavioural trends where appropriate. For example, we excluded reports of one or two birds very late in the season when numbers of the species in question were normally reported in tens or hundreds. In addition, it was not always possible to extract the latest record in a particular season, as sightings may have been reported for every month of the year. In these cases data for the particular year were omitted. Instances of this nature were infrequent but may be interesting to explore in the future as they suggest that some winter migrants may be beginning to remain in Ireland year-round and in the future may become residents, providing that environmental conditions and food supply were suitable.

Identifying appropriate data sets by which to examine the influence of climate change on bird migration patterns poses many challenges (Knudsen et al., 2011), and in the absence of long-term dedicated monitoring networks we may seek alternative means by which to explore such relationships (Donnelly, Yu & Liu, 2014). Some of the limitations of the data, used in the current study include the lack of population-level data, and it would not be appropriate to extrapolate the current trends to a broader geographic area or to a population level. Data representing the tails of a distribution, such as last departure or first arrival, are limited in their usefulness for the reasons just mentioned but nonetheless have been shown to be weakly representative of population size and distribution (Sparks, Roberts & Crick, 2001; Stirnemann et al., 2012). In addition, sampling effort may influence early or late sightings and therefore may introduce bias into the trends (Sparks, Huber & Tryjanowski, 2008). However, we found the number of contributors to the reports to be fairly consistent over time, with an average of 76 per year and with many of the names being repeated over the 24-year period, which would certainly help reduce any such bias. The timing of spring departure of all categories (diver (2), waterfowl (11) and wader (11)) of winter visiting waterbirds examined in this 24-year dataset revealed that the majority (16 out of 24) of species showed a tendency to remain in Ireland later in the season over the time period examined but the trend varied between species. The fact that only very few of the trends were statistically significant is more than likely due, at least in part, to the short length of the time series. Reports of significant trends usually have 30 or more years of data (Cotton, 2003; Miller-Rushing et al., 2008; Donnelly et al., 2009; Lehikoinen & Jaatinen, 2011; Stirnemann et al., 2012). However, the records used here were restricted to the time period 1980-2003. The reason(s) these birds may be staying around longer is unclear but (as mentioned above) may result in some of these birds becoming resident in Ireland. Interestingly, the delayed trend was not observed for all species; eight species, including two categories (waterfowl (3) and wader (5)), showed a tendency to depart earlier. The earlier trend for Whooper Swan, although not statistically significant, was in agreement with a previous study in which daily counts (1972–2008) of this species revealed earlier spring departure from a wintering site in the southern part of Ireland (Kilcolman Wildlfowl Refuge, Co. Cork) (Stirnemann et al., 2012). The early departure was indirectly influenced by an increase in February temperature at the site, which had a direct impact on the food supply. Grass yield (food supply) was greatest in years when February temperatures were warmer than average, thus allowing birds to get in condition and depart early. In addition, competition for resources in the wintering grounds also significantly influenced the timing of departure (Stirnemann et al., 2012). The larger the population, the earlier the departure. It would be interesting to examine whether the birds remaining later in the season are also arriving later in autumn and if there is a shift in the overall timing of arrival and departure of these species.

A study carried out by Lehikoinen & Jaatinen (2011) over a 31-year period (1979–2009) examined daily counts of autumn departure in northern Europe of a number of waterfowl including 6 of the species examined in the current study. Even though the methods of data collection were very different between the 2 studies, some interesting observations could be made. For example, the beginning of autumn departure from Finland was significantly delayed for Eurasian Wigeon, while the departure in spring for the same species in Ireland also tended to be delayed (P = 0.0864). In other words, these waterfowl were found to be staying (in Finland) later in autumn and leaving (from Ireland) later in spring. Greylag Goose, Pintain and Goldeneye at both locations showed similar tendencies. These comparisons suggest a possible shift in the timing of residence in European wintering grounds and highlight the need to examine the entire annual migration of migrant birds rather than focusing on spring or autumn in isolation. However, more detailed data would be required to confirm these patterns.

The tendency towards early arrival of both (only the Sandwich Tern was statistically significant) spring/summer migrants examined in this study was similar to that reported for a number of sub-Saharan migrant visitors to the same region of Ireland (Donnelly et al., 2009). Nine (seven of which were statistically significant) out of eleven passerine species examined revealed an advance in the timing of arrival (1969–1999) which was strongly correlated with average March temperature. Given the different habitats occupied by the passerines and waterbirds, it is nonetheless interesting that a similar pattern of earlier arrival was revealed. Furthermore, the Manx Shearwater is a long-distance migrant wintering at sea in the South Atlantic whereas the Sandwich Tern winters in southern Europe and Africa. Even though these birds travel very different distances, with earliest observed sightings reported in different numbers (626 ± 1,801 (Manx Shearwater) vs 2 ± 2 (Sandwich Tern)) and on average arrive in Ireland more than a month apart their earlier arrival tendency is of interest. It would be useful to examine trends of a range of long- and short-distance species to determine if their arrival followed a similar pattern.

The departure of the winter visitors was influenced by rising spring temperature, especially in March. Both negative (Red-throated Diver and Northern Pintail) and positive (Whooper Swan) correlations were revealed. The majority of winter visiting birds showed positive correlations between the timing of departure and March temperature; although this was not always statistically significant, it was somewhat unexpected. The Whooper Swan tended to depart earlier over the time period (1980–2003), and a previous study (Stirnemann et al., 2012) reported the earlier departure to be negatively correlated with February temperature. However, in this study departure was also negatively but not statistically significantly correlated with February temperature but was positively (statistically significantly) correlated with March temperature. One reason for this may be that March temperature, in reality, has little impact but just happened to be correlated significantly. However, in contrast, the Greylag Goose revealed a non-significant trend towards delayed departure which was significantly correlated with March temperature. When examining departure response to average spring temperature, some birds departed earlier when spring temperature was warmer and most species showed a similar tendency. Sensitivity to warming temperatures varied between species. A 1 °C increase in spring temperature revealed, as expected, a range of responses: from a delay of more than 10 days/1 °C for Northern Pintail to little or no response for other species such as the Greylag Goose. These contrasting trends highlight the complexity of the relationship between bird departure and temperature, the species specific nature of the relationship, and the need to explore a wider range of temperature drivers in order to explain observed trends.

A northward shift in the population centre of gravity of a number of European waterbirds, including Goldeneye, in response to rising winter temperature has resulted in a decrease in population numbers in Ireland and an increase further north in Scandinavia (Lehikoinen et al., 2013). Furthermore, Crowe et al. (2008) reported a general decline in the total number of waterbirds wintering in Ireland, with waders exhibiting the greatest decrease although some species showed an increasing trend. Range shifts and population fluctuations such as these may in future influence observations of arrival and departure if the population ceases to use the habitat where observations are being recorded. In addition, both intra- and inter-specific competition on the breeding and wintering grounds will be impacted by changes in species’ relative abundance (Lehikoinen et al., 2013).

The winter NAO influences environmental conditions early in spring and when in a positive phase food supply and habitat suitability for migrant birds are generally optimized over a relatively large geographical area (Hüppop & Hüppop, 2003). This contrasts with the spring temperature, which represents a more restricted and local effect. The timing of departure of the winter visitors showed stronger correlation with the winter North Atlantic Oscillation than the timing of arrival of the spring visitors. However, this may reflect the low number of spring visitors examined in the study. In general, during a positive NAO phase, the timing of departure was earlier in the year but there were also exceptions, indicating a high level of variability in response. The NAO has been reported to have a strong negative correlation with the arrival of spring migrants (passerines and non-passerines) in many locations in Europe (Žalakevičius et al., 2009; Jonzén, Lindén & Ergon, 2006; Vähätalo et al., 2004; Hüppop & Hüppop, 2003; Forchhammer, Post & Stenseth, 2002), including Ireland (Donnelly et al., 2009) but there have also been reports of delays (Jonzén, Lindén & Ergon, 2006) and no influence (Cotton, 2003). Therefore, it was not surprising to see variation between species in their response to such a large scale circulation patterns. Interestingly, the NAO and local spring temperature more often than not (23 cases out of 26) resulted in the same directional response of spring departure and arrival, suggesting that both large scale and local weather patterns have a similar influence on bird migration.

Conclusions

We have clearly demonstrated a strong relationship between climate and changes in both spring departure and arrival of a number of migratory waterbirds to the eastern region of Ireland over a 24-year period. As with many previous reports, the trends were found to be species specific, as some birds advanced while others delayed the timing of departure. Furthermore, in some years we observed sightings reported in every month, indicating the potential for these migrants to become residents in Ireland. Although we expected to find the majority of birds leaving earlier in response to warming temperatures, we found the opposite, which may be related to the short time series available. This result, combined with reports (from Finland) of species remaining longer in their breeding grounds, suggests that the duration of stay in Ireland may not be changing but may be shifting to later in the season. The length of stay of these birds requires further investigation both within Ireland and across Europe as a whole in order to track potential shifts in migratory patterns at a more regional scale. A shift in the timing of migration will have implications for competition for resources and territory at both the wintering and breeding grounds.

The authors would like to express their gratitude to the volunteers and staff who contributed data to the BirdWatch Ireland East Coast Bird Reports. In addition, we would like to acknowledge Met Éireann for supplying the temperature data and Eileen Diskin for collecting data on contributors.

Additional Information and Declarations

Competing Interests

Author Contributions

The authors declare there are no competing interests.

Alison Donnelly conceived and designed the experiments, performed the experiments, analyzed the data, contributed reagents/materials/analysis tools, wrote the paper, prepared figures and/or tables, reviewed drafts of the paper.

Heather Geyer contributed analysis and writing to earlier drafts.

Rong Yu analyzed the data, contributed reagents/materials/analysis tools, reviewed drafts of the paper.

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
