# Peer review of "Changes in the timing of departure and arrival of Irish migrant waterbirds"

_PeerJ, doi:10.7717/peerj.726_

## Round 0.1 · original submission · Major Revisions

This paper study changes of waterbirds across years and its relation with climate variations. Both reviewers suggest major changes in methods and statistical analysis. It is necessary to follow their valuable suggestions before the validity of the findings can be evaluated

·

Basic reporting

Species data and local temperature data are not sufficiently referenced for the reader to establish how to find them. While all the data used in the correlations with temperature and NAOI are included, the time series' used in the trend testing (Mann-Kendall tests) are not available in the ms and not easily referenced on any public repository. Please refer to PeerJ's data sharing policies for more information.

Experimental design

The collection of species observations as part of the Irish East Coast Bird Reports is unclear - quantitative data on the number of observers, their effort and their experience should be reported. The extent of the study area should be indicated and how well the two local weather stations represent local temperatures in the study areas should be described.

Validity of the findings

As per basic reporting (see above) the data should be made available in conformity with PeerJ's policy.

I have strong reservations about the analytical approach followed. There are 26 species, for each there are computed 8 p-values (1 associated with M-K, and the seven correlations with different months/indices – see table 2) – leading to 208 tests in total. Since p value of <0.05 (5%) is chosen as the significance level, at least 10 of these tests (0.05 * 78) are significant purely by chance. By quick examination of Tables 1 & 2 – it seems ~26 of these tests are significant – thus almost half of these may be spurious! Although there are arguments against it in many cases (e.g. Moran 2003, Oikos 100 (2): 403-405) even a simple sequential Bonferroni correction should be warranted, although preferably something more powerful – see Moran (2003) and later improvements. Without this, incorrect deductions are being made over the significance of many of the trends in the data. I offer some ideas for alternative analytical approaches in General comments below.

In addition to this, in the results and discussion (for specific instances, see General Comments, below) in several places, non-significant results are reported as trends. This is misleading, as in fact the absence of statistical significance in these cases means we cannot really distinguish this level of effect from background variability - in other words these cannot be justified as being any different from a lack of trend. Thus, the majority of species actually show no trend with climate or year even though the authors report otherwise (e.g. L. 216-218, L. 321-323 and several other places - see General Comments below).

In consequence, without use of appropriate multiple test correction factors or a more appropriate analytical framework, it is very unclear as to what conclusions can be drawn from the current results with any confidence.

Also, there are potential problems or implications of how the data were processed and which species were included (for instance, the Sandwich tern, with an average peak count of 2 individuals does not provide any useful information about population-level phenology) or which years or data points were excluded (e.g. the way outliers were identified appeared largely subjective and large annual variation in peak count sizes may complicate these) - see General Comments below.

Finally, it is unclear the degree to which observations of arrival and departure are a compound function of movement and/or phenology (as discussed on L. 303-305) – this really limits the ability of such observations to tell us something concrete about phenology in a population context.

Additional comments

Thank you for the opportunity to review your interesting work. I found the manuscript generally well written and it described an interesting analysis of changes in the observations of arrival/departure of waterbirds in Eastern Ireland with respect to year, temperature and NAOI.

I have described my main reservations with the manuscript as it currently stands in the sections above (chiefly the validity of the analytical approach and the conclusions drawn from it) and provide detailed comments that I hope will facilitate revisions to strengthen the results and conclusions below.

Please feel free to contact me if you have any questions regarding my comments. I hope you find them useful.

Sincerely,

Steve Oswald
[email protected]

Title – suggest adding “ in relation to climate”

Missing postcode for Trinity College

L. 11 – natural systems is rather vague. Do you mean biotic or abiotic. It appears from the following sentence that you mean species responses (in particular birds, since this is what the second cited source addresses).

L. 13-14 again, unsure what local level changes you are discussing – first reference in this sentence suggests that of birds but second reference is so broad it could be changes in climate or weather that are region-specific.

L.21-36 I like this paragraph but am unsure of the purpose. The link from the opening paragraph suggests that you are discussing natural variability as indexed by NAOI but at the end of the current paragraph you discuss NAOI changing in response to anthropogenic activities – thus it is no longer the purely naturally-occurring oscillation alluded to at the beginning of this paragraph. I think it needs to be clear how you view NAOI here as this shapes your paper.

L. 40-41 could do with a general, supporting reference about evolution of migration strategies.

L. 48 again, natural systems is too vague here – does this include birds, their food, resources, weather, climate etc….?

L. 48 “It is inevitable…” Not so. It is highly likely, it could certainly make a valid hypothesis, but I don’t believe it to be inevitable that migratory birds will be affected (again it is vague as to what types of effects you are talking about). For instance, if they are not currently highly constrained in their migrations (e.g. there is plenty of time and resources for migration, a change may not necessarily impact them).

L.49 be specific – what type of temperature – “mean surface temperature” – is this annual temperature, summer temperature, a 30 year mean?

L. 52-54 – it would be good to give an idea of what type of long-distance migrants were studied here – I’m assuming not waterbirds but that should be made clear as it reflects the previous studies upon which the current work is founded.

L. 56 I think Rubolini and colleagues should be formatted as Rubolini et al., but you should check the journal specifications.

L. 62 – insert a comma before “emphasizing”. Also it would be nice to know in a bit more detail why the results of the studies may be at odds with one another.

L. 68 Donnelly et al. in 2006 should probably be Donnelly et al. (2006)

L. 68 – the relevance of Donnelly et al. (2006) here is questionable. The ms is about birds, waterbirds in particular, the phenology of which should not really be related directly to tree budburst. Suggest removal and rewording. Also, I see the point here, to verify that climate change is taking place in Ireland (although this probably is unnecessary, given the general introduction) but should be moved to where temperature changes in Europe are discussed. The introduction is quite long and could benefit with some trimming, particularly here.

L. 77-79 – belongs in the previous paragraph but could probably be removed as I would contest that it is widely understood.

L. 83-86 – it would be nice to see more formal statements and hypotheses that are being tested here. For instance at the beginning of the Introduction it seemed as though this study was going to test the relative importance of natural variation (as indexed by NAOI) vs anthropogenic forcing (as indexed by temperature change). Although this perhaps isn’t valid (as stated by the authors NAOI may be being anthropogenically forced) it would help here to state hypotheses to show the reasoning behind comparing local-scale effects with large-scale effects – one that comes to mind is that the temperature change measurement is a mean (directional) change whereas the NAOI implicitly more cyclical.

L. 86-88 – this is more a sentence for the Discussion or Abstract and doesn’t belong here

L. 92-93 – an introduction – unnecessary here

L. 93-95 – no citation is provided for these data and they are not included as an appendix. I believe to comply with PeerJ policy these raw data should be open-access – if this is not possible, for example special permission was provided to use these data, then citations or weblinks and contact information for access should be provided.

L. 96 – how many birdwatchers? Were these professionals or citizen science data? What was the range of experience of these personnel. Again, a weblink may suffice to answer these questions.

L. 97 – the extent of the East Coast region needs to be specified as well as an indication of the effort – how many personnel? Were observations made every day?

L. 97-98 “earliest dates” should perhaps be “earliest reported dates” – same for “latest dates”

L. 103-106 – rather than raw numbers – what proportion of the mean counts did this represent – e.g. 2-5%? - in theory the proportions should be similar for rare or common species for the records to be truly comparable.

L. 106-108. If a species was recorded in all months of the year this would suggest that even though the breeding population migrates southward in the winter it would be replaced by wintering birds that breed north of Ireland. I’m wondering if including these species is valid. If in one year there are birds year-round but in the next there are none – it is possible that they simply were not detected (what is the likelihood of this – see comment for L. 97). If they are indeed absent, if it is a small population it might be that they wintered on the west coast not the east coast and this is largely a result of chance rather than a real effect of climate. Also, you perhaps are not necessarily measuring the change in phenology of a specific migrant population but the combined movement of a number of different populations – I’m not sure what repercussions this has (if, e.g., date has retreated perhaps the breeding/wintering populations have in fact moved north to be replaced by ones that used to breed in southern Ireland, wintered further south and thus had further to travel so arrived later…). I think it needs to be made clear that we are not necessarily talking about birds that breed/winter in the study area, but migrant species in which some individuals are present as breeders in the summer but some presumably pass through the study area.

L. 115 – reference Table 1, but also provide some reference for the reports from which these data were extracted.

L. 115-117 Although 15 years of data appear to be a good number of data points it needs to be clearer on what basis this was considered to be statistically “sufficient”. Perhaps citing another published study that used this cut-off. In theory, the length of the required data-set to explore a trend depends on the observed variability within it. There is another deeper implication here. Many of the waterbird species studied may live to be 15 years of age, thus it becomes less clear whether any changes are a result of changes in behaviors of individual birds or changes in population-level phenology as a result of populations being composed of different individuals.

L. 117-119 I am presuming that the 14 excluded did not meet the discussed requirement of 15 years of data – but this should be stated clearly. Also, it would be useful to have a supplementary information table listing all 40 species considered.

L. 121-127 – please give lat/longs for the two stations and also a Figure showing the entire study area and how these two locations adequately represent prevailing climate within this study area. Also, as you did for NAOI below, give the website links for these data.

L. 135 provide reference for MULTMK/PARTMK

L. 133-141 I have reservations about the analytical approach, not on the basis of an incorrect choice of test – the use of the Mann-Kendall is certainly an appropriate first step, however in the application of multiple individual tests. For instance, there are 26 species, for each there are computed 8 p-values (1 associated with M-K, and the seven correlations with different months/indices – see table 2) – leading to 208 tests in total. Since p value of <0.05 (5%) is chosen as the significance level at least 10 of these tests (0.05 * 78) are significant purely by chance. By quick examination of Tables 1 & 2 – it seems ~26 of these tests are significant – thus almost half of these may be spurious! Although there are arguments against it in many cases (e.g. Moran 2003, Oikos 100 (2): 403-405) even a simple sequential Bonferroni correction should be warranted, although preferably something more powerful – see Moran (2003) and later improvements. Without this, incorrect deductions are being made over the significance of many of the trends in the data.

I understand the concern over non-parametric approaches to trend detection, but powerful multivariate methods (e.g. GAMMs) do exist whereby one analysis per dataset can be undertaken, with data on species being grouped by a random factor to account for pseudoreplication. An alternative approach, to preserve the analytical power of parametric methods is also available (see Oswald et al. 2012 Methods in Ecology and Evolution, 3(6): 1073-1077.) although this would largely depend on the predictability of changes in climate – e.g. if a non-linear, polytonic response is expected.

L. 145-148 – it is unclear how Fig 1 is constructed as it shows only one time series – were averages made across both sites? – is this also how the data used in correlations were generated as an average across both weather stations? If so, should both stations be expected to be equally representative of the study area? All this should be stated explicitly in the methods.

L. 147 – Rather than significance perhaps the R-squared value should be reported to show the importance of the trend. Note that it is unclear how the “significance” of this trend was determined. The reporting of a slope of 0.04 deg C suggests a regression. However, care should be taken with regressions on time-series’ as much of the predictive power of the regression may be due to temporal autocorrelation and thus the significance values are greatly inflated.

L. 150-152 this should be in the Methods

L. 150-161. Apart from the problems of multiple testing on significance levels as discussed above, I find this paragraph to be a bit misleading by reporting that dates became “progressively later for the majority of species”. Only two of these M-K statistics were significant thus the other 14, although the statistic was positive, could quite likely have been a result of chance effects (since they were non-significant) and thus not a real change but within the realms of variability. This is also the case for L. 190-191.

L. 162-169 and Table 1 – note that in many cases the standard deviations are close to, or even exceed, the average peak numbers of each species. Although counts are Poisson distributed, and thus tend to be positively skewed, and so perhaps medians and quartiles might be better measures of central tendancy, such high standard deviations suggest that in many years numbers were fairly low and in some numbers were fairly high. This not only affects the designation of when to exclude records of a species (see L. 103-106) but also the probabilities for which one might expect to observe the first or last birds of the season. Therefore, earliest or latest dates may be strongly correlated with peak counts – this should be tested for and reported. If this is indeed so, in some years changes in arrival/departure reflect distribution within and outside the study area, more than changes in arrival phenology (or simply arrival phenology in a small region of Ireland that does not necessarily correspond to the area used by a particular population).

Given the small sizes of the populations of some species in the study area (e.g. Sandwich tern: 2+-2) changes in phenology in these cases (even though significant L.161) are likely to be simply the behavioral decisions of individuals or chance rather than true population effects. I believe that species with fewer than 50 during peak counts should probably be excluded from this paper for that reason.

L. 196-202. I think “sensitivity” is probably the wrong word as this suggests a direct effect on the species but when reporting correlations we are not necessarily looking at direct cause-and-effect. Put a different way, this provides some indication that temperature may be an important driver but equally, something else related to temperature might be the causal effect.

L. 201 – is this 16.5 days per 1deg C as for the Grey Plover? Also, the use of days/1 deg C is a slope – not available from a Pearson correlation - and must have been calculated by either regression or simple slope equation y/x. However, I can’t see this stated anywhere in the methods. Note also that the Sanderling trend was non-significant, indicating it is not really a trend or significantly different from 0 days/deg C – ie this should be no effect as for the Greylag Goose.

L 204-205 – remove from results.

L. 204-213 As per comments on previous non-significant correlations – it doesn’t matter if these are negative or positive they should be treated as no different to a zero trend rather than being reported as a trend. Also be careful – NAO is the North Atlantic Oscillation as a phenomenon, NAOI is the index which is what you were analyzing (e.g. L. 212 NAO should be NAOI)

L. 210 – I think perhaps this should refer to Fig 4 – since Fig 2 is on temperature effects not NAOI.

L. 216-218. Given my current misgivings with the multiple analytical approach, I cannot agree that the results show clear evidence for changes in timing. I would imagine some of these are indeed real effects but this cannot be easily determined as the analysis currently stands. Also, “a number of waterbirds” is a bit misleading – how many is this? – certainly not most of them – as only a few show significant trends. In fact, if you wanted to play devil’s advocate, you could say that the results show that most of the waterbirds analyzed showed no clear trend over a 24 year period.

L. 223-225 As indicated in the methods, the systematic and rigorous approach needs to be explained, justified and quantified.

L. 221-235 This is very qualitative discussion of some of the limitations of the data and does not give the reader much understanding of how this might impact the results quantitatively. Much better would be to provide some indication of how often each of these limitations was a problem – what proportion of the dataset was excluded as outliers? In what proportion of instances were birds reported every month?

L. 223-225 – see my comment above about the fact that given the ranges of these birds, perhaps the population moves outside the study area and so what you are getting is differences in behavior between different populations rather than changes in a specific population or subpopulation.

L. 236-239 – in the majority of cases these changes, as indicated by the non-significant M-K tests were not distinguishable from inter-annual variation and thus were not really changes but more sampling variability. Changes should only be reported for the 7 significant cases.

L. 254-266 It is not clear how your results compare to Lehikoinen and Jaatinen (2012). They show 4/6 of the same species showing advancement in arrival and leaving but you don’t specify how that compares to your results here – for instance the Greylag Goose showed no change in your study (Table 1).

L. 267 This is misleading here as you only have two species (of which the Sandwich tern has too few individuals to be analyzed – see above) and thus that you have one species that shows a similar trend could be purely by chance – especially because this trend is non-significant (Table 1).

L. 287-297. This suggests that perhaps the current analytical approach is not the most suitable. That there is a trend towards warming that is not related to departure date but that birds leave earlier in warmer springs suggests that the overall correlative approach between year and departure date is not the best approach. Instead the correlation with temperature suggests a real response and the fact that increasingly early breeding is not seen in the yearly time-series suggests that there is much variability in annual temperatures not that there is no response in phenology to climate.

L. 293-297 – this is true but not unexpected. The migratory behavior and energetic demands of the species are very different and largely reflects their size and the availability and energetic value of their prey.

L. 303-305 – thus observations of arrival and departure are an unknown function of movement and phenology – this really limits the ability of such observations to tell us something concrete about phenology in a population context.

L. 308-319 problems with multiple significance testing aside, only 5 of the 26 species show significant relationships with NAOI (Table 2; 3 are negative, 2 are positive), thus L. 308-310 is misleading and the current study shows that responses may be in either direction. Incidentally, in these cases, NAOI showed often a stronger correlation than local temperature. It would be good here to discuss the general mechanism through which these two indices might impact phenology – in theory ,local temperature gives an indication of food availability in the study area whereas NAOI perhaps gives a better overview of conditions outside the study area that might affect speed of migration.

L. 321-322 – Even without the potential problems with significance and trend interpretation, you cannot make the assertion that climate is driving the phenology with correlative studies – simply that it there is a trend which is consistent with climate as a potential driver for such change.

L. 326-329 – this is a nice idea and should be the focus of this manuscript but it is currently not well supported as only very few of your species show significant relationships – I would contend with the current results that the majority show no such responses.

Figure 1-4 Note all figures report slopes and statistics from linear regressions but it is not stated anywhere in the manuscript that these regressions were performed and how temporal autocorrelation was accounted for.

·

Basic reporting

In this study the authors investigate whether migration phenology of different waterbird species are changing across years, and whether these phenological changes are related to climate variations, specifically to variations in spring temperature and the NAO index. As stated by the authors, changes in migration timings might have important impacts on waterbird populations, but evidence for a relationship between climate and migration phenology is altogether few and inconsistent among species. Therefore this study might be a useful contribution to the topic. The manuscript is generally well written, although some key points should be better addressed (see sections below).

Experimental design

I suggest that some major revisions are made. My main concern regards the use of latest and earliest dates as measures of departure of wintering birds and arrival of breeding birds, respectively. Although these dates are used in other studies and might be reasonable indicators of migration phenology, a number of methodological issues must be addressed (e.g. Sparks et al., 2001). The authors themselves mention some of these issues, but do not address them. The main criticism to this method (as also acknowledged by the authors; ln 101-102) is that latest and earliest dates are measured in one individual bird, thus it might not represent the whole population. This issue can be partly resolved if the surveying effort is constant and standardized across years. Thus the authors must provide more detail on the waterbird surveys: what was (i) the periodicity of the field visits and (ii) the spatial and time extent of the survey (each year)?; (iii) were visited sites and recording effort the same across years? Overall, were the recording methods standardized across years? The representative of these measures might be even more affected if the phenological distribution is thin-tailed, i.e. if the proportion of latest/earliest birds compared to the median is too few (in other words, if these latest/earliest birds are outliers). In addition, population size might also affect migration phenology (as also recognised by the authors; ln 250-251, 303-304); however, its effect is not measured. This relationship (population size vs. latest/earliest dates) can be shown as supplementary information or perhaps included in regression models together with temperature and the NAO index (see also below). Knudsen et al. (2011; Appendix 1) summarize the downsides of this method. (see also General Comments for the Author)

Validity of the findings

The issues mentioned in the previous section ("Experimental Design") should be addressed in order to fully appreciate the validity of the findings. With regard to the interpretation of results, the authors should be more focused on significant patterns, rather than spending so much effort in discussing non-significant directional tendencies (i.e. negative or positive relationships not supported by the data). In fact, the majority of wintering species did not show significant variation across years, while one of the two breeding species showed significant variation. I am not saying that non-significant relationships should not be reported; only that the discussion should be more focused on differences among species (why the direction of the tendency varies among species, and why the majority of species do not show any tendencies?). For example, short- and long-distance migrants might be differently affected by climate warming (as mentioned in the Introduction), which could explain (or not) some of the differences between species. (see also General Comments for the Author)

Additional comments

Specific comments

Abstract
- Following from the comment above, the abstract should be more focused in significant patterns. For example, the authors state that "... whereas Purple Sandpiper would remain just over 2 weeks longer in response to a 1ºC increase in spring temperature", but this relationship is not significant and thus it should not be interpreted as a real pattern.
- "... local temperature had a stronger influence than large scale circulation patterns": I do not find this result in the manuscript. To investigate the relative strength of effects, the authors should perform multiple regression analysis with standardized predictors in order to compare regression coefficients. Or if the aim is at estimating the relative importance of predictors, then variation partitioning should be performed. Multiple regression has also the virtue of limiting type I errors, which can be a serious issue when independently testing pairs of variables (as is the case in this study).

Introduction
ln 18-20: move this last sentence to the beginning of the paragraph.
ln 86: "any patterns": change to "phenological patterns"

Methods
- ln 92-93: delete sentence, as this information was already provided in the Introduction.
- ln 95-97: please provide more details on the surveying method (see the first major comment above)
- ln 97-99: specify that the earliest date corresponds to the date in which the first individual of a breeding species was observed, and the latest date corresponds to the date in which the last individual of a wintering species was observed. This results in one date value per species per year, right?
- ln 104, 105: can "8-10" and "2-3" individuals be given in proportions (of the whole population). Individual numbers do not take into account fluctuations in population size across years. I also suggest showing these results (population size across years), perhaps as supplementary information.
- ln 109-111: can a low number of birds (e.g. <10) be considered a migratory population? Couldn't they be dispersing individuals?
- ln 139-141: correlation results are showed in Table 2. However, in Figures 3 and 4 regression analyses are presented. The details for these analyses should be provided in this section ("Statistical analysis").

Results
- ln 145-148: the methods for these results and Figure 1 (average spring temperature as a function of years) are not provided in the Methods section: are these the results of Mann-Kendall test also?
- ln 152-161 and throughout Results: avoid emphasis on directional tendencies (positive or negative) of non-significant patterns.
- ln 162-169: I suggest to move this paragraph to the beginning of the section.
- ln 171: a latest date of 30 Sept. is a bit odd. How do you know this bird departed for "spring" migration? Couldn't this bird be arriving to winter? Please explain.
ln 197-202: explain this analysis in the Methods
ln 204-205: delete this sentence

Discussion
- ln 223 ("given the systematic and rigorous manner used to generate these data"): this is what authors should explain in the Methods. Provide details as to support the systematic and rigorous surveys.
- ln 227-232: this is important information that should be in the Methods. Refer what "outlier" birds were excluded from the analysis and why. Could the wintering bird allegedly departing in September be an outlier?
- ln 250-251 ("The larger the population the earlier the departure"): I suggest to check this relationship, as it might affect the relationship between migration timing and climate warming. One way of doing it is to perform multiple regression analysis in which temperature, NAO and population size are included as predictor variables (be sure to check the error distribution, as N=24 might be too short for 3 predictor variables). (this is also useful to estimate the partial effects of each predictor)
- ln 303-305: please show population fluctuation in a supplementary figure
- ln 317-318: Because NAO and spring temperature resulted in the "same directional response" of migration timings, both variables might show covariance, which also supports the use of multiple regression.

References

Knudsen, E., Lindén, A., Both, C., Jonzén, N., Pulido, F., Saino, N., Sutherland, W.J., Bach, L.A., Coppack, T., Ergon, T., Gienapp, P., Gill, J.A., Gordo, O., Hedenström, A., Lehikoinen, E., Marra, P.P., Møller, A.P., Nilsson, A.L.K., Péron, G., Ranta, E., Rubolini, D., Sparks, T.H., Spina, F., Studds, C.E., Sæther, S.A., Tryjanowski, P. and Stenseth, N.C. 2011 Challenging claims in the study of migratory birds and climate change. Biological Reviews 86, 928-946.

Sparks, T. H., Roberts, D. R. and Crick, H. Q. P. 2001. What is the value of first arrival dates of spring migrants in phenology? Avian Ecology and Behaviour 7, 75–85.

---

## Round 0.2 · accepted · Accept

The manuscript have been improved including the Reviewers suggestions. Data have been referenced and clarified, references have been included and results have been restructured. Congratulations and Merry Xmas!